# Anorexia Nervosa in Adolescence: Parental Narratives Explore Causes and Responsibilities

**DOI:** 10.3390/ijerph20054075

**Published:** 2023-02-24

**Authors:** Luna Carpinelli, Meike Watzlawik

**Affiliations:** 1Department of Medicine, Surgery and Dentistry, Baronissi Campus, University of Salerno, 84081 Baronissi, Italy; 2Department of Development, Education and Culture, Faculty of Psychology, Campus Tempelhof, Sigmund Freud University Berlin, 12101 Berlin, Germany

**Keywords:** anorexia nervosa, subjective illness theory, eating-disorder-focused family therapy, positioning analysis, agency

## Abstract

Background: Anorexia nervosa (AN) is a serious mental disorder with a multifactorial etiopathogenesis, adolescent girls being especially vulnerable. Parents can be a resource and occasionally a burden when their children suffer from AN; thus, parents play a key role in recovery. This study focused on parental illness theories of AN and how parents negotiate their responsibilities. Methods: To gain insights into this dynamic, 14 parents (11 mothers, 3 fathers) of adolescent girls were interviewed. Qualitative content analysis was used to provide an overview of the parents’ assumed causes for their children’s AN. We also looked for systematic differences in the assumed causes among different groups of parents (e.g., high versus low self-efficacy). A microgenetic positioning analysis of two mother–father dyads provided further insight into how they viewed the development of AN in their daughters. Results: The analysis stressed the overall helplessness of parents and their strong need to understand what was going on. Parents differed in stressing internal and external causes, which influenced whether they felt responsible and how much they felt in control and able to help. Conclusions: Analysing the variability and dynamics shown can support therapists, especially those working systemically to change the narratives within families for better therapy compliance and outcomes.

## 1. Introduction

Currently, there is a widespread misconception that eating disorders are a lifestyle choice. According to the Italian National Institute of Health (INIH) [1], eating disorders (EDs) are serious and often fatal diseases. The Diagnostic and Statistical Manual of Mental Disorders (DSM-V) [2] reports a 12-month prevalence of 0.4 per cent for AN and 1–1.5 per cent for BN in young women and represents a growing public health problem. Furthermore, recent decades have seen a progressive lowering of the age of onset of EDs and, with no respite during the COVID-19 pandemic, an increase of approximately 30% in diagnoses [3].

AN is a serious mental disorder with a multifactorial etiopathogenesis caused by a combination of biological, sociocultural, and psychological factors [4]. Researchers do not agree about a potential genetic influence on AN [5], but a genetic component seems likely [6]. In particular, AN presents a special risk for adolescent girls and young adult women with an approximate ratio of 10:1 for females versus males [7]. In general, non-binary and trans* youth seem to be at greater risk of experiencing eating disorder symptoms than their cis-gendered peers [8], but further research is needed here.

In addition to gender, attachment style is another risk factor. Attachment represents the way in which a person relates with and binds to others, while adjusting behaviours, thoughts, and emotions accordingly [9]. The safe and insecure attachment styles are the two that are most often differentiated. Secure attachment is considered the most adaptive, featuring a good balance of trust, commitment, and exploration so that relationships can be shaped according to mutual needs. Insecure attachment styles, on the other hand, are many, yet all are characterized by maladaptiveness [10,11] and may be found frequently among individuals with AN. The most important influence on attachment style is the interaction with primary caregivers. Approximately 70% of people with AN also display an insecure attachment style that avoids or concerns [12]. In adults, the avoidant–insecure attachment style is characterized by a fear or anxiety of being emotionally involved in interpersonal relationships, and by the adoption of a defensive position against others to prevent disappointment or rejection. Conversely, concerned–insecure attachment is characterized by continuous concerns and increased perfectionism in the relationship, which influences strategies of behaviour and adjustment.

Over the past few decades, the scientific literature related to the maintenance factors of EDs (specifically for AN) has focused on different aspects of relationships and family functioning. Often, families of persons with AN may be characterized by dysfunctional relationship styles that may foster an environment of tension and hostility due to numerous conflicts. These family environments and insecure attachment styles support the development of AN, and affect the number and severity of symptoms, the degree of success and compliance with treatments, and the risk of relapse; thus, they deeply hinder progress and the success of therapy [13,14]. In addition, families of patients with AN show worse overall family functioning than control groups. Furthermore, research focusing on caregiving has shown the presence of high levels of expressed emotion and criticism and a lack of positive management of psychological distress in AN patients’ families compared to control groups [15]. That “family functioning” refers to a complex construct also becomes apparent in the review study by Gale et al. [16], which investigated the father–child relationship as a risk or maintenance factor for ED in adolescence: Conflict, communication, sense of protection, psychological control, emotional regulation, and self-esteem seemed to influence the adolescent’s level of autonomy and may consequently have had an impact on maladaptive eating attitudes and psychopathology.

Yet, there are also cases where a person with AN was found within a healthy, balanced family environment, one that featured appreciative communication as well as solid problem-solving skills. In these cases, the family was a protective and not a risk factor. Patients with AN who come from these types of families will adhere more strongly to care, face the therapeutic path in its entirety, and experience a very low risk of relapse [13,14].

In addition, an important factor increasingly associated with family therapy is parental self-efficacy. In fact, in family-based treatment for adolescents with AN, an improvement in parental self-efficacy has been associated with a reduction in adolescents’ eating disorder, anxiety, and depression [17]. However, families involved in treatment for adolescent AN present a pessimistic view in relation to the family’s ability to focus on behaviours that reduce AN symptoms. Therefore, understanding the degree of family flexibility and perceived parental self-efficacy could be important in assessing treatment outcomes [18].

The social context can thus be of central importance when assessing the development of AN. This is confirmed in particular by studies demonstrating that patients with early-onset AN (before the age of 18) of less than three years’ duration will profit from systemic family therapy [19]. Linville and Blow [20] also summarized that “families [including parents, but also siblings and other family members] can be a resource and occasionally a liability for their loved one who is suffering from an eating disorder and can therefore play a key role in their recovery;” the authors therefore also advocated for a systemic treatment of eating disorders. Eating-disorder-focused family therapy seems to be one of the strongest evidence-based treatments for adolescent AN, because its aims include establishing a supportive and non-blaming context [21]. This seems rather challenging, because parents themselves struggle to understand the disease and to determine how they can best support their children. In remarks to interviewers, these parents described how they were facing a “dreadful monster” and a “living nightmare” [22]. Existing qualitative findings from interview studies with parents showed that despite their struggles, parents developed sophisticated explanations about their child’s eating disorder [23].

The aims of this qualitative–quantitative study were to (a) take a closer look at these parental theories of illness and (b) analyse how the way that they talk about their children’s disease reflects their perceived agency in the process of dealing with it.

## 2. Materials and Methods

### 2.1. Procedure

Parents of female AN patients were recruited during the first knowledge meeting of the psycho-education programme organized by a clinic specialized in eating disorders located in Campania region (Southern Italy). Selection criteria were (a) having cared for their daughter for at least six months; and (b) having a daughter with EDI-3 GPM > 70 [24]. Parents (both mothers and fathers) were asked to join the research protocol and leave their contact details. Within 48 h, they were contacted to make an appointment and received an e-mail containing the informed consent form and a link to the platform to participate in the meeting. Interviews lasted 40–45 min on average and were conducted by the first author between November 2021 and March 2022. The interview protocol was constructed ad hoc based on the variables of the study, and two sections were planned: a qualitative section with eight open-ended questions and a quantitative section that consisted of screening tests.

### 2.2. In-Depth Semi-Structured Interviews

The exploratory, in-depth, semi-structured interview consisted of eight questions: (1) “What, in your opinion, is anorexia?”, (2) “Could you tell me about your experience with anorexia?”, (3) “What factors do you think contributed to developing or maintaining this type of eating behaviour?”, (4) “From your point of view, what could be done to prevent this problem?” (5) “You turned to (facility) … about a problem with AN … What motivated you the most to ask for help?”, (6) “How would you describe the type of help/assistance received at the service?”, (7) “Had you looked for other forms of help before coming here? If YES, can you tell me about these previous experiences (why didn’t they go on? What difficulties did you encounter?)”, (8) “Is there anything you would like to add to your image of mental anorexia in general, or to this interview? Other aspects that you think are useful to investigate?”.

These open-ended questions let parents describe their overall experiences, their beliefs about what caused their daughter’s eating disorder, possible strategies to prevent it, and suggestions for improving care.

### 2.3. Additional Measures

The quantitative section featured, among other things, the following standardized tests:
The Parenting Self-Efficacy Scale (PSES), which assesses perceived sense of self-efficacy in relation to parenting skills [25]. It consists of 12 items to be answered on a 4-point Likert scale (from 0 = not at all capable to 3 = very capable). Cronbach’s alpha for the total PSES was 0.85, indicating a moderately high level of internal consistency among scale items. Scores ranged from 0 to 36; higher scores indicated higher levels of self-assessed parental efficacy.The Relationship Questionnaire (RQ), which has four measurable categories of attachment styles: secure (low anxiety and avoidance), fearful (high anxiety and avoidance), preoccupied (high anxiety and low avoidance), and dismissing (low anxiety and high avoidance) [26]. The RQ is a single-item measure consisting of four short paragraphs, each describing a prototypical attachment pattern as it applies to close relationships in adulthood. There were two parts, RQ1 and RQ2. In the first part, RQ1, participants were asked to select a paragraph-long description that best described them, without providing a numerical rating. In the second part, RQ2, participants were asked to rate their agreement with each prototype on a 7-point scale. The highest of the four attachment prototype ratings was then used to classify participants into an attachment category.A 12-item General Functioning Scale (GF-FAD) of the Family Assessment Device (FAD). Each of the 12 items can be used alone or as part of the 60-item FAD, the most widely used assessment device in research on family functioning [27]. A mean for each scale was generated, ranging from 1 to 4, with higher scores indicating greater distress. The cut-off score for the GF-FAD is 2.0. The test has good test–retest reliability and concurrent reliability, and multiple studies found that it was able to discriminate between clinical and control samples [28].

### 2.4. Participants

Overall, 14 parents (11 mothers and 3 fathers) participated in this study. Their daughters were all diagnosed with AN, based on DSM-5 criteria and the Eating Disorders Inventory (EDI-3) test [24]. Daughters (n = 12) had a mean age of 15.9 (SD = 1.4; range |13–17|), an average BMI of 16.8 (SD = 2.1; range |12–19|) at the time of assessment, and a Global Psychological Maladjustment (GPM) >70 (clinical range cut-off).

The parent group had a mean age of 50.57 (SD = 5.84, range |42–64|). The number of household members varied from min 2 to max 5.78.6% who were married or cohabiting, 14.3% who were separated or divorced, and 7.1% who were widowed. For education level, 35.7% had a high school diploma, 28.6% had a bachelor’s degree, and 35.7% had finished middle or primary school. As to employment, 28.6% were independent contractors, 21.4% were permanent employees, 21.4% were housewives, 14.3% were unemployed, 7.1% were temporary employees, and 7.1% were retired. For caregiving duties, 57.1% had been taking care of their anorectic daughters for one to two years, 35.7% for less than a year, and 7.1% had been involved for more than two years. When asked who within the household was mainly responsible for the care of their daughter, 64.3% answered predominantly the mother, 28.6% chose both equally, and 7.1% said predominantly the father. All the parents came from the province of Naples (southern Italy), which is under the territorial jurisdiction of the Local Health Unit where the clinic is located.

From this group of participants, two sets of parents (mothers and fathers) were selected for an additional microgenetic positioning analysis (see below) [29].

### 2.5. Data Analysis

*Quantitative Analyses.* The three scales described above were analysed descriptively using IBM–SPSS software v.23 (SPSS Inc., Chicago, IL, USA) to provide an overview of how the parents assessed their current situation.

*Qualitative Content Analysis* was used to identify the subjective illness theories of all parents. The procedure followed a summarizing, inductive approach to providing an overview of all causes that the parents considered relevant to their child’s illness [30]. With the help of the software MaxQDA, a coding scheme was developed and refined in the recursive process of analysis (communicative validation). The frequency with which the parents mentioned certain explanations were calculated, and formed the basis for the group comparisons via contingency tables and chi-square tests (SPSS).

*Microgenetic Positioning Analysis* is based on the observation that we construct identities via interactions with others, claiming and assigning certain positions to ourselves and to them [28]. In addition to claiming a certain position for oneself (“I am overwhelmed”), positions are assigned to other people (e.g., to the children, as being “vulnerable”) and to the disease (AN) itself (e.g., “the monster”). All acts of positioning (no preselection) were collected in tables for the six interviews (self, others, AN). Afterwards, the results from the mothers were compared with those of the accordant fathers with the help of illustrations (see Figure 1 and Figure 2), highlighting similarities but also contradictions between and within the interviews. Agency, the perceived “power to act”, and its counterpart passivity [31] were described for the different positions when possible. Quality criteria for qualitative research were above all transparency, comprehensibility, and richness of detail, in order to show the dynamics in the parent–child relationship in this case. The two selected cases were therefore analysed close to the text and without pre-selection of certain passages. If passages were paraphrased, this was consented in the team (communicative validation).

## 3. Results

### 3.1. Family Situation from the Parents’ Point of View

*Parental Self-Efficacy (PSE).* The parents’ mean score on the PSE scale was 25.9 (SD = 6.0), determining a medium level of perceived self-efficacy. The group was divided overall into parents with low (values below 20; 21.4%), medium (values between 21 and 30; 42.8%), and very high self-reported self-efficacy (values above 30; 35.7%).

*Parenting Attachment Style (RQ).* Of our sample, 42.9% presented a dismissing attachment style, 21.4% a secure attachment style, while 21.4% were preoccupied, and 14.3% were fearful. Avoidance could thus be observed frequently in this sample, since both the dismissing as well as the fearful group evinced this behaviour. In 21.4% of the parents, anxiety dominated, and only 21.4% were securely attached.

*Family Functioning (GF-FAD).* A total of 42.9% of the sample exceeded the clinical cut-off of the GF-FAD scale (>2.00). This figure is representative of the fact that the family system highlighted by the parents who were interviewed falls into a disabling and dysfunctional type of conduct. Dysfunctional families are simply those that lack the psychological resources for their members to cope with coexistence in an assertive and developmentally favourable manner.

Overall, the low number of secure attachments as well as the low level of family functioning confirm the findings presented in the introduction, but also show that a secure parenting attachment style as well as high parental self-efficacy do not necessarily prevent AN.

### 3.2. Qualitative Content Analysis: Subjective Beliefs about What Causes AN

Table 1 provides an overview of all 18 causes the parents named. The top category, which seven parents saw as important, was condition of the mind/emotional state, describing how AN is not a physical condition, but one that starts in the mind or on an emotional level; only later do physiological symptoms show up. Lack of or low self-esteem was also named by seven parents as the major cause for AN.

*Social media as well as media use* in general was the other top answer that parents gave when asked what had caused their child’s eating disorder. In most cases, parents were referring to the beauty standards displayed on social platforms and the possibility (if not the obligation) to compare oneself with others. Beauty standards in general were named by four parents. In other cases, to the parents, social media represented superficial and non-fulfilling relationships.

Five parents stated that their children had certain character traits that made them vulnerable and prone to develop an eating disorder, or that loneliness/isolation was the starting point. Overall, more internal causes (e.g., feeling distressed, lonely, vulnerable) than problematic behaviours (e.g., social media use, dieting) or external causes (e.g., beauty standards, peer pressure, the pandemic) were assumed to be important. For some children, parents reported that the dieting experience and higher self-awareness/lower self-acceptance correlated with the onset of puberty.

As frequencies indicate, most parents did not give one but several reasons for their child’s eating disorder, while only two explicitly stated that the cause was multi-factorial. However, only two parents indicated that they may have contributed to the condition themselves (parenting style). One person pointed out that family problems (not their own) could also be relevant for eating disorders in general.

Frequencies were too low to systematically assess group differences, and the only striking feature in the data was that all (100%) of the dismissing parents stated that AN was a condition of the mind. No fearful or preoccupied parent repeated this statement, and only one securely attached parent agreed.

### 3.3. Microgenetic Positoning Analysis: Two Sample Case Studies

The two parental dyads chosen for the microgenetic analyses is introduced separately in the following two sections.

In the first dyad, the parents were both 49 years old; both were self-employed. Their 13-year-old daughter was their only child. She had been diagnosed with AN over a year (but less than two years) ago. In the interview, the mother did not mention any family or personal problems but stated that such problems should be addressed to prevent AN in general, because they can be “very strong, very serious family problems”. Whether or not this was the case in her own family was unclear. The father did not mention any family or personal problems either. Neither the father nor the mother spoke about the other as an individual; the other parent appeared only in the form of “we”, on the dyadic level. While both stressed their helplessness on an individual level (low agency), agency was stressed when the parents referred to themselves as a couple (“we did something”). While the mother generally positioned herself as strong and confrontational, the father did not provide any positive self-positions (see Figure 1). 

**Figure 1 ijerph-20-04075-f001:**
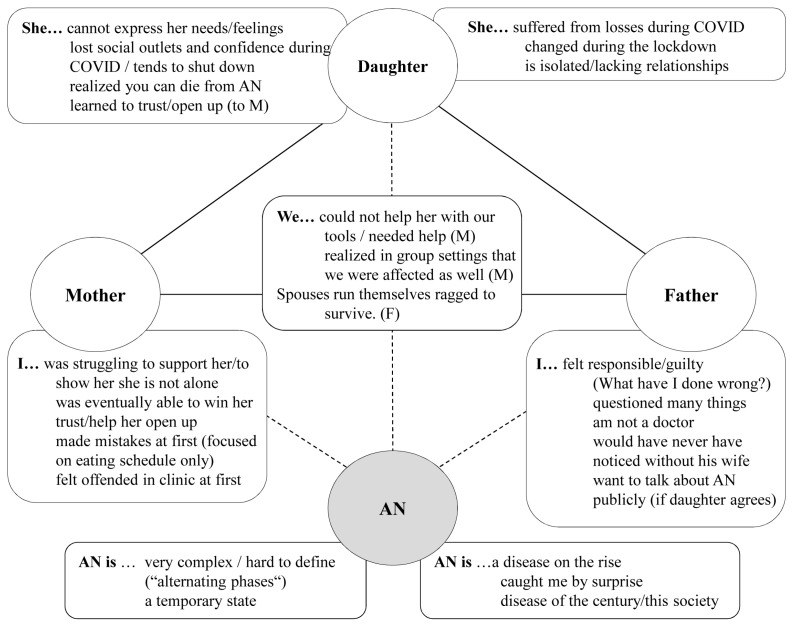
First parental dyad: overview of positions claimed for oneself (mother/father/dyad) and assigned to the daughter, and anorexia nervosa (AN) (M = mother; F = father).

He also stated that he did not know what caused the initial depression that he considered to be the starting point of AN in his daughter. Both positioned their daughter as a (naïve) victim (low agency) but stressed that she was uncooperative and reckless at certain points, suggesting that she could potentially have behaved differently from the parents’ point of view (some agency). When positioning AN itself, the mother used very emotional terms and metaphors (“the monster”); she was painting the image of an enemy that had to be defeated, and for which she needed “the right weapons,” weapons that she did not yet have. The father approached AN on a cognitive level by trying to understand what the problem was, hopefully to be able to solve it—which, from his perspective, was harder than one might assume.

In the second dyad, the mother was 54 years old while the father was 55 years old; both of them worked as independent contractors. Of their two children, the 16-year-old daughter had been diagnosed with AN less than a year before. The parents here agreed on their positions; the mother was the one who noticed that her daughter needed help. Although she was struggling, made mistakes in the beginning, and felt temporarily disempowered in the clinic when she was put on the spot while looking at her own family system, she eventually was able to help her daughter. The interview seemed to demonstrate a difficult process but ultimately a success story of overcoming a temporary state. The mother positioned her daughter as someone who had a difficult time opening up, so that finding a way to communicate openly with her became an even greater success (see Figure 2).

**Figure 2 ijerph-20-04075-f002:**
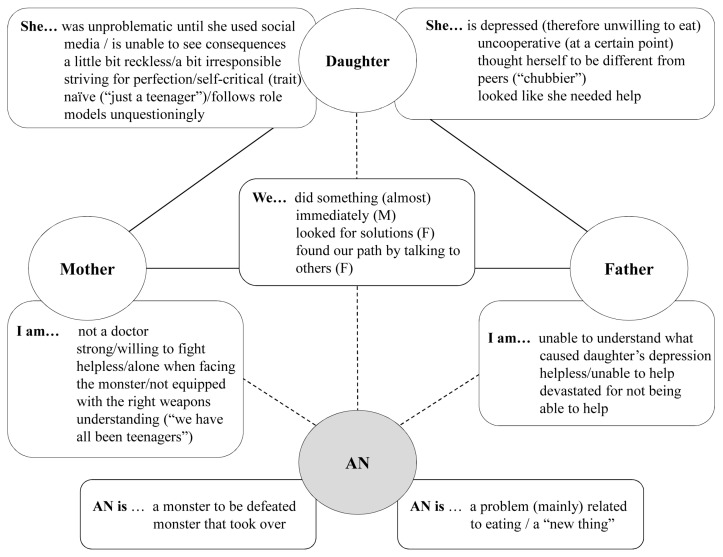
Second parental dyad: overview of positions claimed for oneself (mother/father/dyad) and assigned to the daughter, and anorexia nervosa (AN) (M = mother; F = father).

The father lacked agency throughout when talking about his own family system. Before the onset of the illness, he had not noticed that anything was going wrong, and in the interview, he hardly mentioned or described his daughter. His general statements on society may have been an indirect reference to his own family, in which he was the “provider”, the one who provided the money and security—but, by focusing on that, he had lost sight of more important things:

because generally the parent, especially fathers, especially in an age like today he thinks he gives everything by giving what the children want […] the dress, the trip, the cell phone, the money to go out at night, but I mean… then clearly you find out […] that all of that is not enough and so then it goes back to what I was saying before to a world, to a type of society that goes two hundred kilometers per hour towards material goods leaving out everything else […] a family that is no longer there, in fact (father D2, Q5)

In the beginning, he explicitly stated that he felt guilty and responsible. He also did not talk about the parents as a unit, but declared on a meta-level that spouses nowadays have to run themselves ragged to survive—which might be a justification for not being there enough. With his daughter’s condition improving, he was looking for ways to regain agency by wanting to talk about AN—which he called “the disease of the century”—publicly to inform parents and help them support their children in similar situations.

## 4. Discussion

The aim of the present study was to investigate family perceptions regarding AN especially in relation to their own experience, with the help of two case studies supported by qualitative and quantitative data. The existing literature on parental care of children with anorexia [32,33] showed that parents, in managing the interaction with their child, developed negotiation processes in which they attempted to mediate between their own feelings of guilt and frustration and their striving to understand the illness in order to adapt to the child’s care [34]. The same could be shown here in detail with, e.g., in the second case study, the clashing role expectations of the father and mother. It is evident that being confronted with one’s own responsibilities, and potentially maladaptive behaviours, as well as fluctuating stages in one’s own child (from help-seeking to resistant) is a challenge for most parents who must simultaneously negotiate their own roles within the family and outside (e.g., at work).

The quantitative data of this study confirmed current findings by showing that the secure parenting attachment style is not the one most frequently observed among parents of girls diagnosed with AN; most of these parents displayed a dismissing attachment style that is characterized by low anxiety and high avoidance [35]. In addition, 42.9% of our sample exceeded the clinical cut-off of the Family Functioning Scale (GF-FAD > 2.00). These findings might not only be interpreted as a cause but also as an effect of AN as well, one that arises in the form of helplessness, role confusion, shame and guilt—a conclusion that the qualitative findings support, which will be discussed further below.

The assumed cause for AN seems to have an impact of what kind of solutions the parents strive for. All dismissive parents stated that AN was a condition of the mind; the cause was, thus, assumed to be *inside* their children, which, again, might be difficult for the parents to detect and, therefore, for them to influence. If this is the major explanation, avoidance becomes understandable in that it circumvents frustration and disappointment in the treatment process [12]. Surprisingly, self-efficacy varied widely between parents but did not vary systematically with attachment styles or family functioning. This may indicate that self-efficacy displays a temporary state the parents are in—one that is influenced by multiple factors [36]. 

The qualitative data of this study can help shed light on the complexity of the healing process, in which subjective disease theories are a key element. The interviewed parents widely agreed that AN had its onset in a bad emotional/psychological state for which low self-esteem was a major feature. Social media, through which cultural beauty standards are reinforced and displayed, was a central danger in the eyes of the parents, along with social isolation and a lack of relationships with peers [37]. None of the parents mentioned possible positive effects that social media might have; for example, helping young people stay in touch with friends during the pandemic. While for some families, the pandemic itself proved to be a reinforcing factor in AN, for others, it was the major catalyst that triggered the child’s eating disorder in the first place.

Internal as well as external causes were blamed by the parents, along with problematic behaviour by their children, who were positioned as victims: naïve, resistant, and striving for perfection. Only three parents reflected on their own roles in the emergence of the disease. The microgenetic analysis of the second dyad, for example, included a turning point that helped parents overcome this avoidance: group sessions at the clinic made the parents rethink their positions because they noticed that others were in the same boat. Guilt and helplessness apparently let parents focus on what was easiest to handle; for example, controlling the calorie intake of their daughters. The mother in the second dyad, nevertheless, evaluated this focus as deceiving and wrong in retrospect and asserted that building a relationship and team effort with her daughter was of much greater importance.

The microgenetic analyses also showed that family narratives and identity construction as well as roles taken can vary widely between family members. In both case studies, the mothers actively faced a threat while the fathers found themselves in a less active role. While in the first dyad, the parents eventually seemed to act as team—even as the father tried to find comfort in intellectual explanations, and the mother armed up against a monster—the second dyad had a clear role distribution, the mother as the daughter’s supporter and the father as provider, looking at the processes from the outside. He found comfort in the idea of changing society on a greater scale—even if that was not possible at home.

Reviewing the explanations given by the parents for why eating disorders develop, and the results concerning the daughters in the microgenetic positioning analyses, it seems important to recognize that AN is caused by inner and outer circumstances. Instead of taking control because the AN sufferer is considered to be incapable, parental striving for empowering a child and working as a team holds the possibility of a promising onset, as the second dyad 2 showed, without losing sight of the fact that parents might need help finding fulfilling roles for themselves in both a professional and a private context. 

## 5. Conclusions

The initial and sometimes ongoing helplessness of parents when a child is diagnosed with AN became strongly apparent in this study. Parents, nevertheless, deal with the situation very differently, so that their personal disease theories as well as the identity work associated with them must be considered in therapy to best address the different needs of all family members. The limitation of this study was the small sample size in at the quantitative data. This was the reason we only used it for descriptive purposes. Additionally, the sample was predominantly made up of mothers, who are mainly responsible for the care of their daughters. To date, the involvement of fathers is increasingly required, especially in family support therapies [38].

However, we can hypothesize the presence of certain factors that may influence resistance to paternal involvement. For example, it is possible that paternal involvement reflects a more specific family functioning. It is possible that families with a disorganized relational style, a chaotic parent–child relationship, or a poor marital relationship are less likely to participate in care pathways [38].

Future research should recruit fathers to better identify any differences in caregiving. Inferential statistical analyses would need to be conducted with bigger samples to show correlational as well as mediating/moderating effects. Additionally, longitudinal data would be of interest for describing changes over time, with the help of quantitative as well as qualitative data (change of positionings). On the one hand, it would be possible to discover more precisely what the maintaining factors of AN are, and on the other hand, the so-called risk factors could be elaborated: how exactly do families function and how exactly does dysfunctionality manifest itself over time? Indeed, many factors seemed to be out of the control of the parents; thus, it is essential to work on those aspects that one can actually influence. This includes the child–parent relationship as well as the parent–parent relationship (if applicable), as it is also a central subsystem in the family. This work also includes the question of who one is oneself, who one was and wants to be, as well as the discovery of others and their specific needs and how these needs can best be supported.

## Figures and Tables

**Table 1 ijerph-20-04075-t001:** Overview of assumed causes and frequencies (parental illness theories; N = 14): light grey = internal causes; grey = problematic behaviour; dark grey = external causes.

Nr.	Code	Sample Answers	N
1	Condition of the mind/ emotional state	“Start[s] at the mental level, at the level of the psyche certainly. It’s a mental block, however, that leads to all the consequences that come later.”	7
2	Lack of/low self-esteem	“A loss of self-confidence ... a time when she questioned herself a lot” “She has low self-esteem, that she does not like herself”	7
3	(Social) Media use	“Television—bombarding the brain with models” “Instagram, Facebook and many other things that lead them to see themselves as even more different from the world”	6
4	Question of character	“Can also come from the character”/“Shyness”	5
5	Loneliness/isolation	“The loneliness”/“Has been distant and therefore not in relationships”	4
6	Pandemic/lockdown	“Lockdown exacerbated the symptoms and everything happened from there.”	4
7	Diets/rejection of food	“Adolescence started around age 13, started with a diet, in her own way”	4
8	Beauty standards	“Today’s world is based on images, on perfection.”	4
9	Society in general	“Induced by the dictates of today’s society”	3
10	Traumatic events	“Some love disappointments”/“Childhood trauma”	3
11	Social context	“Influenced by the social context”	2
12	Parenting style *	“I am a very anxious person—maybe even that … I was on her all the time ....”	2
13	Onset of puberty	“Since adolescence, […] she can hardly look at herself in the mirror”	2
13	Symptom of other disorder	“Depression”/“Personality disorder”	2
14	Multi-factorial	“There are so many facets and so many reasons why a girl […] gets anorexia.”	2
15	Family problems *	“Problems that exist within families”	1
16	Possibly genetic	“It could be genetic.”	1
17	Bad influence from others	“Someone out of envy put into her brain that she had to lose weight.”	1
18	Boredom	“Boredom”	1

* Parents hint at their own responsibility for the development of AN.

## Data Availability

Not applicable.

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
