# Peer review of "Anorexia Nervosa in Adolescence: Parental Narratives Explore Causes and Responsibilities"

_ijerph, 2023, doi:10.3390/ijerph20054075_

Round 1

Reviewer 1 Report

GENERAL IMPRESSIONS

The present paper deals with a really important topic in EDs psychopathology and treatment. As the authors stated, EDs usually have a strong and negative impact on family functioning, but the family itself can represent a protective factor during recovery, so it is important for psychotherapists to invest in it.

However, the present paper has some problems related to the scopes of the work itself. In fact, it seems not clear which are the research questions: at some points, the authors declear that the goal is to analyze the social and familiar factors which affect the development of AN (lines 70-71), but, the experimental part seems to be dedicated rather to a description of the parents’perception about EDs and their causes. 

Moreover, the authors anticipated that the results of the present work will be useful for psychotherapists during treatment planning, but in the discussions this argument is never more deepened.

Then, we reccomend some revisions:

ABSTRACT

·       From line 10: when the authors introduce the article, they use the present simple (“focuses”); however, during the following lines they leave the present simple and start to use the past simple for all the verbs.

INTRODUCTION

·       In the first part of the introduction (lines 33-61) the auhtors present a lot of precentages, in order to describe various prevalences in EDs (gender prevalences, age prevalences, and so on). Since the scope of the article is not related to the socio-demographic features of EDs, we think that it could be more useful if the authors dedicated the very first lines of the paper to the description of the most important features of EDs which would be fundamental for the paper itself;

·       Line 46: there is a typo (“AM” instead of “AN”);

·       Line 65: it would be more correct if the abbreviation “EDs” would be presented in the singular form;

·       Lines 66-67: while presenting the different etiological factors in EDs, the authors rapidly change topic and start to talk about the medical complications and the suicide risk in AN. We recommend to delete the sentence in lines 66-67;

·       Lines 79-81: here the authors state that the attachment style does not rise from a vacuum. It would be really important if they discussed here about the influence of the parental attachment style on the child’s one style. In fact, in the experimental part, the authors investigate the parental attachment styles, but they never discuss about the specific role that they have in EDs/AN pathology. From line 86, the authors explain the role of familiy in EDs’ development, but they never deepen the dynamics which connect parental attitudes (or attachment styles) and specific attachment styles in children, failing to convey in what way these styles affect pathogenesis of EDs;

·       Line 87: maybe, the sentence “Families of people with AN are, in many cases, dysfunctional” could sound a little judgmental. Please consider to re-write the sentence by using other words;

·       Line 111: the sentence is declined in past simple, while the other ones in the paragraph are in present simple.

MATERIALS AND METHODS

·       Line 120: the authors describe the sample as a group of ED patients’ parents; however, in lines 184-186 it appears that all the children have received a diagnosis of AN. It is not clear whether the inclusion criteria included other EDs, and the resulting AN-only sample is a consequence of the recruiting, or if the recruitment in fact targeted AN population only. It would be useful to explicitly present inclusion/exclusion criteria in the methods section;

·       Regarding the procedure, the authors state that they recruited the parents after some meetings of psychoeducation. Since the scope of the paper is to investigate how parents perceive AN illness, it would have been better if the authors had interviewed the parents before the psychoeducational intervention, because in this way we cannot exclude an influence of the intervention on the parents’ views, making this sample not necessarily representative of the general population;

·       Since the semi-structured interview has been entirely designed by the authors, it would be helpful if they gave a detailed description of all the items included;

·       The authors included the Parenting Self-Efficacy Scale (PSES) in the battery, but they did not previously introduced the specific role of parental self-efficacy in AN psychopathology, thus not explicitly providing a rationale for including this questionnaire nor explaining how does it relate to the primary outcomes of the study.

DATA ANALYSIS

·       At lines 201-202, the authors mention some statistic procedures for which results and comment seem to be absent. Results should be included and appropriately addressed in the relevant sections.

·       The authors introduce the microgenetic positioning analysis without providing references regarding the methodology specifics or the validity/reliability of the methodology. These information are critical to appropriately evaluate the methodological robustness of this part of the study.

RESULTS

·       Overall, the language used to report the study results does not always seem appropriate and implies some judgmental stances by the authors (i.e. questioning the honesty of the participants - lines 273-274), which look out of place in a scientific article. If considerations regarding potential biases in the sample should be made, these should be reported in the discussions and study limits sections, not in the results paragraph.

·       Line 313: the alternate use of the singular and plural form may sound a little confusing.

DISCUSSION

·       The declared scope of the paper may be too general. Since there is a wide research on family features of AN population, the work itself would be more original f it had a more specific target and scope. At the current state this work does not appear neither a replication confirmatory study nor it is clear which new input it provides to the field. The scientific impact and clinical relevance of the work should be more explicitly stated by the authors;

·       During the whole discussion, there are too few citations which supports the paper’s results. They should be integrated with more references to published scientific works;

·       Lines 339-343: in these lines the authors state that dismissive parents consider AN as a psychological, mental condition. Then the authors deduce that it is for this reason that those parents develop a detached attitude towards the ED. Since the attachment style is a variable that usually precedes the development of AN (and also influences its development), this different way of thinking should be supported by inserting some references;

·       The paper focuses on families’ perception about AN. However, the major part of the sample is composed by only mothers, even if the 78.6% of them were married (so, the family is not composed only by one mother and children – if these were separated from their husband it is not reported). This should be cited as a limit of the study, since in this way the authors have collected data only from a part of each family (we know that, particularly in case of psychological disturbances, the two parents often do not agree about their perception of the problem).

Author Response

GENERAL IMPRESSIONS

The present paper deals with a really important topic in EDs psychopathology and treatment. As the authors stated, EDs usually have a strong and negative impact on family functioning, but the family itself can represent a protective factor during recovery, so it is important for psychotherapists to invest in it.

However, the present paper has some problems related to the scopes of the work itself. In fact, it seems not clear which are the research questions: at some points, the authors declear that the goal is to analyze the social and familiar factors which affect the development of AN (lines 70-71), but, the experimental part seems to be dedicated rather to a description of the parents’perception about EDs and their causes. 

Moreover, the authors anticipated that the results of the present work will be useful for psychotherapists during treatment planning, but in the discussions this argument is never more deepened.

Authors R1: We thank you for your feedback and suggestions for improving our work. All changes are marked in yellow in the text. 

Then, we reccomend some revisions:

ABSTRACT

  • From line 10: when the authors introduce the article, they use the present simple (“focuses”); however, during the following lines they leave the present simple and start to use the past simple for all the verbs.

Authors R2: Done

INTRODUCTION

  • In the first part of the introduction (lines 33-61) the auhtors present a lot of precentages, in order to describe various prevalences in EDs (gender prevalences, age prevalences, and so on). Since the scope of the article is not related to the socio-demographic features of EDs, we think that it could be more useful if the authors dedicated the very first lines of the paper to the description of the most important features of EDs which would be fundamental for the paper itself;

Authors R2: We reshaped the introduction by focusing on the research topic.

  • Line 46: there is a typo (“AM” instead of “AN”);

Authors R3: Done

  • Line 65: it would be more correct if the abbreviation “EDs” would be presented in the singular form;

Authors R4: Done

  • Lines 66-67: while presenting the different etiological factors in EDs, the authors rapidly change topic and start to talk about the medical complications and the suicide risk in AN. We recommend to delete the sentence in lines 66-67;

Authors R5: Done

  • Lines 79-81: here the authors state that the attachment style does not rise from a vacuum. It would be really important if they discussed here about the influence of the parental attachment style on the child’s one style. In fact, in the experimental part, the authors investigate the parental attachment styles, but they never discuss about the specific role that they have in EDs/AN pathology. From line 86, the authors explain the role of familiy in EDs’ development, but they never deepen the dynamics which connect parental attitudes (or attachment styles) and specific attachment styles in children, failing to convey in what way these styles affect pathogenesis of EDs;

Authors R6: We have included the specifications in both the introduction and discussions. 

  • Line 87: maybe, the sentence “Families of people with AN are, in many cases, dysfunctional” could sound a little judgmental. Please consider to re-write the sentence by using other words;

Authors R8: Remodulated

  • Line 111: the sentence is declined in past simple, while the other ones in the paragraph are in present simple.

Authors R9: Remodulated

MATERIALS AND METHODS

  • Line 120: the authors describe the sample as a group of ED patients’ parents; however, in lines 184-186 it appears that all the children have received a diagnosis of AN. It is not clear whether the inclusion criteria included other EDs, and the resulting AN-only sample is a consequence of the recruiting, or if the recruitment in fact targeted AN population only. It would be useful to explicitly present inclusion/exclusion criteria in the methods section;

Authors R10: Remodulated and included inclusion criteria.

  • Regarding the procedure, the authors state that they recruited the parents after some meetings of psychoeducation. Since the scope of the paper is to investigate how parents perceive AN illness, it would have been better if the authors had interviewed the parents before the psychoeducational intervention, because in this way we cannot exclude an influence of the intervention on the parents’ views, making this sample not necessarily representative of the general population;

Authors R11: We specified that the recruitment took place at the first cognitive meeting of the psycho-education programme so as to show no influence. 

  • Since the semi-structured interview has been entirely designed by the authors, it would be helpful if they gave a detailed description of all the items included;

Authors R12: included

  • The authors included the Parenting Self-Efficacy Scale (PSES) in the battery, but they did not previously introduced the specific role of parental self-efficacy in AN psychopathology, thus not explicitly providing a rationale for including this questionnaire nor explaining how does it relate to the primary outcomes of the study.

Authors R13: Included 

DATA ANALYSIS

  • At lines 201-202, the authors mention some statistic procedures for which results and comment seem to be absent. Results should be included and appropriately addressed in the relevant sections.

Authors R14: Remodulated

  • The authors introduce the microgenetic positioning analysis without providing references regarding the methodology specifics or the validity/reliability of the methodology. These information are critical to appropriately evaluate the methodological robustness of this part of the study.

Authors R15: Including references. The authors also specify that: 

In qualitative research, reliability is not a valid criteria for quality. Especially, if the construct itself (here identity) is considered to be a phenomenon in flux. If a person told the same story twice (exact same), one would assume that it has been learned by heart (for whatever reason) and does not represent actual experience. What will stay the same, if e.g. a father told the story about his daughter twice, are the overall ambivalences, although he might, over time, develop a story in which he is more agentic than in the first. Which in itself would be interesting for an identity researcher.

RESULTS

  • Overall, the language used to report the study results does not always seem appropriate and implies some judgmental stances by the authors (i.e. questioning the honesty of the participants - lines 273-274), which look out of place in a scientific article. If considerations regarding potential biases in the sample should be made, these should be reported in the discussions and study limits sections, not in the results paragraph.

Authors R16: remodulated

  • Line 313: the alternate use of the singular and plural form may sound a little confusing.

Authors R17: remodulated

DISCUSSION

  • The declared scope of the paper may be too general. Since there is a wide research on family features of AN population, the work itself would be more original f it had a more specific target and scope. At the current state this work does not appear neither a replication confirmatory study nor it is clear which new input it provides to the field. The scientific impact and clinical relevance of the work should be more explicitly stated by the authors;

Authors R18: We have reshaped the discussion and conclusion section to be more specific in both the research focus and limitations of the study.

  • During the whole discussion, there are too few citations which supports the paper’s results. They should be integrated with more references to published scientific works;

Authors R19: included more references

  • Lines 339-343: in these lines the authors state that dismissive parents consider AN as a psychological, mental condition. Then the authors deduce that it is for this reason that those parents develop a detached attitude towards the ED. Since the attachment style is a variable that usually precedes the development of AN (and also influences its development), this different way of thinking should be supported by inserting some references;

Authors R20: included more references

  • The paper focuses on families’ perception about AN. However, the major part of the sample is composed by only mothers, even if the 78.6% of them were married (so, the family is not composed only by one mother and children – if these were separated from their husband it is not reported). This should be cited as a limit of the study, since in this way the authors have collected data only from a part of each family (we know that, particularly in case of psychological disturbances, the two parents often do not agree about their perception of the problem).

Authors R21: inserted this point as limitation in conclusions  

Reviewer 2 Report

The article entitled Anorexia nervosa in adolescence: Parental narratives explore causes and responsibilities presents a relevant and current theme, which is extremely necessary to contribute to reducing the prevalence of anorexia in adolescents. The article is well developed and presents robust and conclusive results.

Only in the conclusions is it necessary to present possibilities for future studies, gaps in the literature that this research could not identify or answer.

Author Response

The article entitled Anorexia nervosa in adolescence: Parental narratives explore causes and responsibilities presents a relevant and current theme, which is extremely necessary to contribute to reducing the prevalence of anorexia in adolescents. The article is well developed and presents robust and conclusive results.

Authors R1: Thank you for your time. It is crucial for us to implement knowledge in this area of research in order to organise better support strategies for caregivers. 

Only in the conclusions is it necessary to present possibilities for future studies, gaps in the literature that this research could not identify or answer.

Authors R2: We have included limitations and future research ideas in the "Conclusions" section. 

Reviewer 3 Report

A very interesting topic, however there some points need improvement.

1. Authors mention that sample is too small. In addittion to this,  it refers only to a part of Italy, that' s why their conclusions can not be generalized. Sample contains mostly mothers. It would be more interesting if they had answers from both parents even if they were divorced. I think authors should mention this

2. In the discussion authors do not refer to previous studies and do not compare their results with these studies. They repeat again their results and try to explain. This section needs improvement.

3. They should add as a limitation of their study, that the majority were mothers and that their study refers only to a small part of Italy. Their study would be more interesting if they had answers of both parents. 

4. Their introduction and materials-methods is too extended compared to discussion

5. Finally, abstract should have the form " introduction- materials methods- results- conclusion"

Author Response

A very interesting topic, however there some points need improvement.

Authors R1: Thanks for the feedback and suggestions!

  1. Authors mention that sample is too small. In addittion to this,  it refers only to a part of Italy, that' s why their conclusions can not be generalized. Sample contains mostly mothers. It would be more interesting if they had answers from both parents even if they were divorced. I think authors should mention this

Authors R2: We reshaped both the discussion and conclusion paragraphs on these points by inserting the limitations of the study. 

2. In the discussion authors do not refer to previous studies and do not compare their results with these studies. They repeat again their results and try to explain. This section needs improvement.

Authors R3: Remodulated

3. They should add as a limitation of their study, that the majority were mothers and that their study refers only to a small part of Italy. Their study would be more interesting if they had answers of both parents. 

Auhtors R4: Inserted this point as limitation and future progress of the study. 

4. Their introduction and materials-methods is too extended compared to discussion

Authors R5: Remodulated both paragraphs

5. Finally, abstract should have the form " introduction- materials methods- results- conclusion"

Authors R6: Remodulated 

Round 2

Reviewer 1 Report

GENERAL IMPRESSIONS: we thank the authors for implementing the recommended reviews. Nevertheless, there are some which are missing, or that are not completely satisfying.Especially, we suggest to review the following points:

INTRODUCTION:

• Lines 29-31: the subject of the sentence changes from “DSM” to “EDs”, but the word “EDs” is not present. In this way the sentence itself may result unclear; • Line 63: unclear use of English (a person with AN sufferer);

MATERIALS AND METHODS:

• For the first inclusion criteria (which relates with the previous treatments of the patients) a better description is needed. Specifically, the authors should specify if the previous treatment included some meetings with the patients’ families;  • The authors included the Parenting Self-Efficacy Scale (PSES) in the battery, but they did not previously introduced the specific role of parental self-efficacy in AN psychopathology, thus not explicitly providing a rationale for including this questionnaire nor explaining how does it relate to the primary outcomes of the study;

RESULTS

• Overall, the language used to report the study results does not always seem appropriate and implies some judgmental stances by the authors (i.e. questioning the honesty of the participants - lines 255-256), which look out of place in a scientific article. If considerations regarding potential biases in the sample should be made, these should be reported in the discussions and study limits sections, not in the results paragraph; • Line 297: the alternate use of the singular and plural form may sound a little confusing;

DISCUSSIONS:

• In the very first lines of the discussion, the authors states that the scope of the article is to deepen the inner family processes accompanying the treatment of an AN patient. However, the work itself seems to be more focused on the family perceptions about AN than on the changes of these perceptions as a result of therapy. We suggest to clarify the scopes better, in order to give the paper itself a more consistent form; • Lines 328-332: in these lines the authors state that dismissive parents consider AN as a psychological, mental condition. Then the authors deduce that it is for this reason that those parents develop a detached attitude towards the ED. Since the attachment style is a variable that usually precedes the development of AN (and also influences its development), this different way of thinking should be supported by inserting some references.

Author Response

GENERAL IMPRESSIONS: we thank the authors for implementing the recommended reviews. Nevertheless, there are some which are missing, or that are not completely satisfying.Especially, we suggest to review the following points:

Authors: Thank you for your feedback and suggestions. 

INTRODUCTION:

• Lines 29-31: the subject of the sentence changes from “DSM” to “EDs”, but the word “EDs” is not present. In this way the sentence itself may result unclear; • Line 63: unclear use of English (“a person with AN sufferer”);

Authors: remoduladed 

MATERIALS AND METHODS:

• For the first inclusion criteria (which relates with the previous treatments of the patients) a better description is needed. Specifically, the authors should specify if the previous treatment included some meetings with the patients’ families;  • The authors included the Parenting Self-Efficacy Scale (PSES) in the battery, but they did not previously introduced the specific role of parental self-efficacy in AN psychopathology, thus not explicitly providing a rationale for including this questionnaire nor explaining how does it relate to the primary outcomes of the study;

Authors: implemented

RESULTS

• Overall, the language used to report the study results does not always seem appropriate and implies some judgmental stances by the authors (i.e. questioning the honesty of the participants - lines 255-256), which look out of place in a scientific article. If considerations regarding potential biases in the sample should be made, these should be reported in the discussions and study limits sections, not in the results paragraph; • Line 297: the alternate use of the singular and plural form may sound a little confusing;

Authors: The analysis of the results from our work is descriptive and not interpretative. It is not possible to make a judgement as this would not be consistent with the purpose of our contribution. The explicit statements emerged from the parents' narratives.

DISCUSSIONS:

• In the very first lines of the discussion, the authors states that the scope of the article is to deepen the inner family processes accompanying the treatment of an AN patient. However, the work itself seems to be more focused on the family perceptions about AN than on the changes of these perceptions as a result of therapy. We suggest to clarify the scopes better, in order to give the paper itself a more consistent form; • Lines 328-332: in these lines the authors state that dismissive parents consider AN as a psychological, mental condition. Then the authors deduce that it is for this reason that those parents develop a detached attitude towards the ED. Since the attachment style is a variable that usually precedes the development of AN (and also influences its development), this different way of thinking should be supported by inserting some references.

Authors: Remoduladed the paragraph

Reviewer 3 Report

Although authors made a great effort to improve their manuscript, still some points need improvement

1. English language need improvement, some phrases such as " the literature tell us..." , "in this article...."  for example need rephrasing

2. Again the main problem still remains. Authors do not have answers from both parents. Although they mention that for the two dyads they have a different section, it would be more informative if they had answers for all. And if this was impossible they should explain why. 

Author Response

Although authors made a great effort to improve their manuscript, still some points need improvement

Authors: thank you for you important suggestions. 

  1. English language need improvement, some phrases such as " the literature tell us..." , "in this article...."  for example need rephrasing

Authors: remoduladed

2. Again the main problem still remains. Authors do not have answers from both parents. Although they mention that for the two dyads they have a different section, it would be more informative if they had answers for all. And if this was impossible they should explain why. 

Authors: motivated and inserted in the paragraph as "Limitations"